# Emergent order in epithelial sheets by interplay of cell divisions and cell fate regulation

Philip Greulich[1,2]*

**1** School of Mathematical Sciences, University of Southampton, Southampton, United Kingdom, **2** Institute for Life Sciences, University of Southampton, Southampton, United Kingdom

\* P.S.Greulich@soton.ac.uk

## Abstract

The fate choices of stem cells between self-renewal and differentiation are often tightly regulated by *juxtacrine* (cell-cell contact) signalling. Here, we assess how the interplay between cell division, cell fate choices, and juxtacrine signalling can affect the macroscopic ordering of cell types in self-renewing epithelial sheets, by studying a simple spatial cell fate model with cells being arranged on a 2D lattice. We show in this model that if cells commit to their fate directly upon cell division, macroscopic patches of cells of the same type emerge, if at least a small proportion of divisions are symmetric, except if signalling interactions are laterally inhibiting. In contrast, if cells are first 'licensed' to differentiate, yet retaining the possibility to return to their naive state, macroscopic order only emerges if the signalling strength exceeds a critical threshold: if then the signalling interactions are laterally inducing, macroscopic patches emerge as well. Lateral inhibition, on the other hand, can in that case generate periodic patterns of alternating cell types (checkerboard pattern), yet only if the proportion of symmetric divisions is sufficiently low. These results can be understood theoretically by an analogy to phase transitions in spin systems known from statistical physics.

**Data Availability Statement:** Programming code for the computer simulations made for this work is available at https://github.com/philipgreulich/epith-sheets.

## Author summary

A fundamental question in stem cell biology is how a cell's choice to differentiate or not (cell fate choice), is regulated by communication with other cells in a tissue, and whether these choices are a one-way path or to some degree reversible. However, measuring this in living animals is very difficult and often impossible, since this requires to make videos of cells inside the body with a microscope. Here, we employ a simple mathematical model for the fate choices of stem cells when they are regulated by communication with nearby cells in the tissue. We show that different means of cell fate choice and cell communication can lead to qualitatively different macroscopic features of the spatial arrangement of cell types: large patches, checkerboard patterns, or randomly disordered distributions, depending on the character of cell communication, and whether cell fate is committed at cell division or reversible. Our analysis therefore shows that those aspects of stem cell

**Funding:** This work was funded by the Medical Research Council (https://www.ukri.org/councils/mrc/) New Investigator Research Grant MR/R026610/1 to PG. The funders had no role in study design, data collection and analysis, decision to publish, or preparation of the manuscript.

activity, which are otherwise difficult to measure, can be distinguished by observing spatial arrangements of cell types.

## Introduction

The development of complex tissues requires the appropriate spatial arrangement of cell types. In many organs, cell types are ordered in a certain way, either as regular arrangements, such as hair follicles in skin or crypts and villi in the intestine, or they are clustered into large, yet irregular domains, such as $\beta$-cells in Langerhans islets in the human pancreas [1, 2], prosensory domains in the mammalian inner ear [3], or patches in human epidermis [4, 5]. In other tissues, cell types may be dispersed without apparent order. Understanding the emergence of macroscopic order, be it as regular patterns or irregular domains/patches (see Fig 1), is one of the fundamental questions of developmental biology.

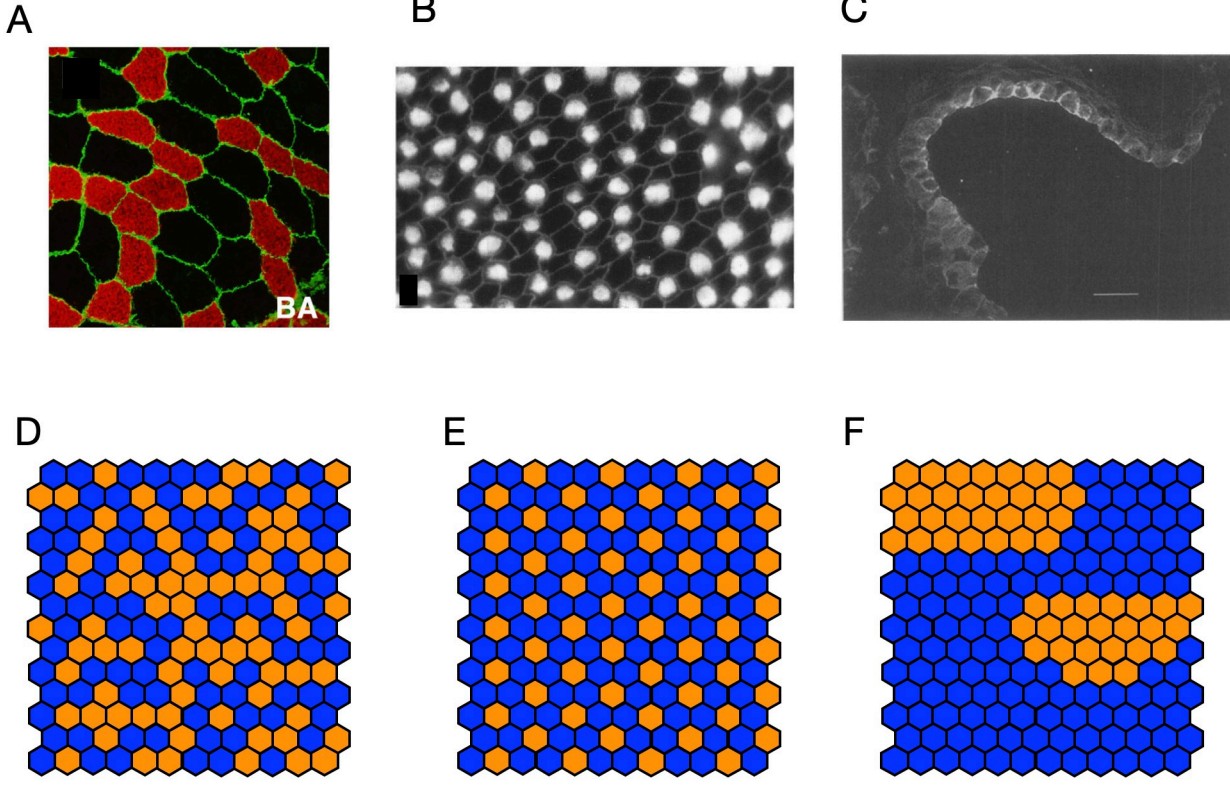

**Fig 1. Fluorescent images of spatial arrangements of cells of two types.** Top row: (A) Muscle cells with 'slow' fibres (red) and 'fast' fibres (black) in human biceps brachii biopsies (Reprinted from [15], on CC-BY license), representing a random arrangement of cell types. (B) Hair (bright) and support cells (dark) in chick basilar papilla (Reprinted from [16], Copyright 1997 Society for Neuroscience), representing a regular, alternating cell type pattern. (C) Integrin expression (bright), marking epidermal stem cells in the basal layer of human epidermis (shown is a 1D section of a 2D epithelial sheet), representing non-random cell type patches (Reprinted from [5], with permission from Portland Press, see also images in [4]); scale bar 50$\mu$m. (Bottom row) Illustrations of qualitative features of spatial cell type arrangements, where blue and orange tiles denote two different cell types in a cell sheet. These correspond to the two cell types in the respective panels above, and also to cell types A and B in the models introduced in the *Model* section). (D) Illustration of a random distribution of two cell types. Random clusters can emerge but they have a fractal structure and the two cell types appear in approximately equal ratios. (E) Periodic pattern (here: with periodicity of one cell length), (F) Irregular large patches. In contrast to a random distribution, cell type clusters have smoother boundaries and single large patches may dominate, so that one cell type occur more often than the other.

Historically, pattern formation in biology has also been a fundamental subject of study in mathematical biology. Motivated by Turing's and Wolpert's seminal works on patterning by long-range morphogen signalling [6–8], partial differential equations have often been employed to model the spatiotemporal dynamics of morphogen signalling and cellular responses in a coarse-grained and deterministic manner. However, a cell's choice to acquire a certain cell type identity (*cell fate choice*) is often regulated by paracrine signalling between neighbouring cells, called *juxtacrine signalling*, and is also subject to some degree of randomness. An example for juxtacrine signalling is the Notch pathway, which can receive signals from neighbouring cells through membrane bound Jagged and Delta-like ligands. This signalling pathway can, depending on circumstances that are not yet entirely understood, either lead to *lateral inhibition* [9–11], when neighbouring cells mutually repress signalling activity and attain preferably opposite cell type identity, or *lateral induction* [3, 11–14], when neighbouring cells mutually activate signalling and prefer equal cell identity. In this case, stochastic agent models that consider randomness and the system at single-cell resolution are more appropriate to study the effect of interactions.

Understanding the mechanisms underlying the emergence of ordered structures in such systems is of paramount importance for tissue engineering and regenerative medicine. Furthermore, this information may also be used to infer the modes of cell fate choice in tissues, also called *self-renewal strategies* in homeostasis. The most commonly employed method to infer self-renewal strategies is by using clonal data from genetic cell lineage tracing assays [17, 18]. However, competing models can, in homeostatic tissues, often not be distinguished based on clonal data [19, 20]. For example, a long-standing question in stem cell biology is whether cells fully commit to their fate at the point of cell division [21], or whether stem cells fluctuate reversibly between states more or less primed ('licensed' [22]) for differentiation, independently of cell division, before finally committing to terminal differentiation [19, 22, 23]. Only intra-vital live imaging has so far, in few tissues, been able to resolve this question [24–26], yet this technique is difficult and expensive, and not feasible in all tissues. Hence, other ways to distinguish self-renewal strategies by using fixed tissue samples would be invaluable. If it is known how different self-renewal strategies generate qualitatively different macroscopic patterns of cell type distributions, which could be observed using appropriate molecular markers in fixed tissues, such a distinction could be made.

To see whether such an approach could be possible for self-renewing epithelial sheets in homeostasis, we will study a simple cell-based model of cell fate choice in a two-dimensional spatial arrangement of cells (a stochastic *cellular automaton* model [27–29]), and we will assess what types of long-range spatial ordering are predicted to emerge for different means of juxtacrine signalling (such as the Notch and its ligands) and self-renewal strategies. Tissues with such a quasi-two-dimensional arrangement of cells are, for example, the basal layers of epidermis and oesophagus, or epithelial (organotypic) cultures, but also other tubular yet flat epithelia, like the mammary gland epithelium, can be approximated by such a spatial arrangement. Cellular automata models have been used in the past to model, for example, the lateral-inhibition effect of Notch-Delta and found that when cells are able to switch between their types, checkerboard patterns of cell types can emerge [10, 30–32] (see also Fig 1B). More generally, it was found that when cell phenotype is determined by reversible genetic switches, a cellular automaton model akin to the Ising model, a paradigmatic lattice model originally developed to understand magnetism [33, 34], can help understand some aspects of cell type arrangements [35, 36]. In the case where Notch acts to mediate lateral induction and in cases where extended cell membrane protrusions can transmit signals beyond nearest cell neighbours, these patterns can have varying lengths of periodicity [37, 38] and exhibit dynamic switching [39]. On the other hand, cellular automata models have also been employed to study the effect of cell

division and cell fate choices under crowding control (but without cell type specific regulation), that is, when every lost cell is replaced by the division of a nearby one [19, 40], which bears resemblence to the voter model of statistical physics [41].

While some works have studied cell fate choices and others cell-type specific (juxtacrine) regulation, so far the direct interplay of both, and its effect on large scale ordering of cell types, has not been studied. Here, we wish to explicitly study how cell division and subsequent fate choices may compete with regulatory cues from the immediate cellular environment, to form large-scale features of spatial cell type arrangements. In particular, we will analyse which features of cell fate regulation and cell fate choice patterns would predict the particular large-scale features of cell type arrangements, as observed in several tissues (see Fig 1). In the future, those predictions about qualitative features of cell type patterns can be compared with data representing the spatial distribution of cell-type specific molecular markers, and thereby mechanisms of juxtacrine signalling and fate choice could be discerned and inferred.

## Models and methods

### Model

To analyse order formation in homeostatic epithelial sheets, we model the interplay between divisions of stem cells, cell fate choices, and juxtacrine signalling between neighbouring cells as a stochastic (Markov) process. We seek to keep this model simple enough to allow theoretical insights and comprehensive understanding, yet sufficiently complete to include the commonly encountered features of signalling, cell fate choice, and lineage hierarchies in homeostatic tissues [19, 21, 42]: We consider the scenario of a unipotent lineage hierarchy, with self-renewing stem cells at the top of the hierarchy, which can differentiate, upon which they leave the epithelial sheet. This is represented as two categories of cells, a self-renewing category $A$, which is not committed and can divide long term, while the other category $B$ comprises cells which are primed ('licensed') or committed to differentiation. Each of these two categories may contain multiple cell types as would be classified by molecular markers or phenotypes, but for notational convenience, we denote those two categories as 'cell types' in the following. Furthermore, we assume cells to be spatially arranged in a square lattice formation, which facilitates the analysis of ordering phenomena, as we can compare it with known stochastic lattice models. While in reality, the spatial arrangement of cells in tissues is more complex, the universal nature of critical phenomena such as macroscopic ordering, suggests that these will qualitatively prevail also in more complex arrangements of cells [43, 44]. Finally, cell division and *fate choice*—that is, the process of cells choosing their cell type identity—are modelled by the combination of two standard models [19, 21], expressed schematically as,

$$A \rightarrow \begin{cases} A + A \\ A + B \quad, \qquad B \rightarrow \emptyset \\ B + B \end{cases} \tag{1}$$

$$A \leftrightarrow B . \tag{2}$$

Here, event (1), left, represents the division of $A$-cells upon which each daughter cell chooses to either remain an $A$-cell or to become a $B$-cell, i.e. fate decisions are coupled to cell division [21]. Event (2), on the other hand, allows cell fate choices to occur independently of cell division [19] and instead of committing immediately, $B$-cells are only 'licensed' to differentiate and retain the potential to return to the stem cell state, $A$ [22]. Finally, event (1), right,

represents the extrusion of $B$-cells from the epithelial sheet (it is assumed that cells continue the differentiation process elsewhere, e.g. in the supra-basal layers of the epithelium, but this is not modelled here). Now, when placing cells in the spatial context, further constraints are introduced. First, we assume that cells can only divide when a neighbouring cell creates space when being extruded from the epithelial sheet. That is, we couple division of an $A$-cell to the synchronous loss of a neighbouring $B$-cell, and vice versa. Hence, only where an $A$-cell is next to a $B$-cell, written as $(A, B)$, the configuration of cells can change: the $B$-cell is extruded, $B \to \emptyset$, which is immediately followed by a division of the $A$-cell, in which one of the daughter cells then occupies the site of the previous $B$-cell. We can express this as,

$$(A, B) \xrightarrow{\lambda \cdot p_A^\lambda} (A, A), \qquad (A, B) \xrightarrow{\lambda \cdot p_B^\lambda} (B, B) \ , \tag{3}$$

where $\lambda$ is the rate at which loss, and coupled to it a symmetric division event, is attempted—while this attempt may not be successful if the chosen neighbour is not of opposite cell type. $p_{A,B}^\lambda$ denotes the probability of fate choice $A$, $B$, of both daughter cells upon symmetric cell division. Here, we only model symmetric division events of the type $A \to A + A$, $A \to B + B$ explicitly. While asymmetric divisions, producing an $A$ and a $B$ cell as daughters, are assumed to occur, they do not change the configuration of cells, since this corresponds to the event $(A, B) \to (A, B)$ (we do not consider events $(A, B) \to (B, A)$ as it is commonly observed that stem cells retain their position upon asymmetric division [24, 45]), and are thus not explicitly modelled. Furthermore, cell fate choice independent of cell division is possible as,

$$A \xrightarrow{\omega \cdot p_B^\omega} B, \qquad B \xrightarrow{\omega \cdot p_A^\omega} A \ , \tag{4}$$

where $p_{A,B}^\omega$ denotes the probability of fate choice $A$, $B$, upon an attempted cell fate choice independent of cell division, which happens at rate $\omega$.

Finally, we consider that juxtacrine (cell-cell) signalling takes place between neighbouring cells, which affects cell fate choice. We model this by allowing the cell fate probabilities $p_{A,B}$ (for simplicity we neglect the superscripts here) to depend on the configuration of neighbouring cell types. In particular, we assume that the fate of a cell on site $i$ depends only on the number of neighbours of type $A$, $n_A^{(i)}$, and the number of neighbours of type $B$, $n_B^{(i)}$ (for an update according to (3), this encompasses all six neighbours of the two sites that are updated). Since in homeostasis, the dynamics of the two cell types must be unbiased and thus symmetric with respect to an exchange of all cell types $A \leftrightarrow B$, the cell fate probabilities must be functions of the difference of neighbouring types $n_i := n_A^{(i)} - n_B^{(i)}$. If $p_A$ is increasing with $n_i$, the excess of neighbouring $A$ cells, this interaction is called *lateral induction*, and if it decreases with $n_i$, it is called *lateral inhibition* [11]. To select appropriate functions $p_{A,B}$, we first note that the competition between the cell types must be neutral for a homeostatic state to prevail, hence we require that $p_{A,B}(-n_i) = 1 - p_{A,B}(n_i)$, which also implies $p_A(n_i = 0) = p_B(n_i = 0) = 1/2$. Furthermore, the probabilities $p_{A,B}$ should, for very large numbers of neighbours of the same type, tend to $p_A \to 1$, $p_B \to 0$ (for lateral induction) or $p_A \to 0$, $p_B \to 1$ (for lateral inhibition) if $n_i \to \infty$ (while the maximum number of neighbours is 4 and 6, respectively, we can in principle extrapolate this function). This asymptotic behaviour suggests a sigmoidal function for $p_{A,B}(n_i)$. We test two types of sigmoidal functions, one representing an exponential approach of the limiting value, modelled as a logistic function, the other one an algebraic approach,

modelled as a Hill function. Since $p_A(n_i = 0) = 1/2$, we therefore choose,

$$p_A^{(log)}(n_i) = \frac{1}{2}\left(1 + \tanh(Jn_i)\right) \quad \text{(logistic)} \tag{5}$$

$$p_A^{(hill)}(n_i) = \frac{1}{2}\left(1 + \frac{Jn_i}{1 + |Jn_i|}\right) \quad \text{(Hill)} , \tag{6}$$

and $p_B(n_i) = 1 - p_A(n_i) = p_A(-n_i)$. In these equations, the parameter $J$ quantifies the strength of the interaction, that is, how much the cell fate probability is affected by neighbouring cells. Note that here we used a symmetrized version of a Michaelis-Menten function (Hill function with Hill exponent 1) to assure the symmetry, as other Hill functions cannot be symmetrized in that way.

In the following, we wish to study whether the mode of cell fate choice affects the spatial patterning of cell type distributions. One fundamental question in stem cell biology is whether cells commit to their fate at the point of cell division, or if this choice occurs independently of cell division and is reversible [19, 22]. To address this question, we consider two model versions. In the first version, cells divide according to events (3), and $B$ cells are assumed to irreversibly *commit* to differentiation (*model C*), i.e. no events according to (4) occur. In the second version, we assume that cell fate can be chosen independently of cell division, in a *reversible* manner (*model R*), i.e. transitions $A \rightarrow B$, $B \rightarrow A$ according to (4) can occur. In both cases, fate regulation by juxtacrine signalling is determined by the functional forms of $p_{A,B}^\lambda(n_i)$ (for model C) and $p_{A,B}^\omega(n_i)$ (for model R), according to (5) and (6). Formally, the two model versions are defined through specific choices of parameter values in the general model, namely,

$$\text{model C}: \quad \omega = 0 \tag{7}$$

$$\text{model R}: \quad p_A^\lambda = p_B^\lambda = 1/2 . \tag{8}$$

where the equality of $p_A^\lambda$ and $p_B^\lambda$ in model R is to ensure homeostasis in the limit $\omega \rightarrow 0$. This means that, effectively, in model C, only $p_{A,B}^\lambda$ is a function of neighbour configurations as in (5) and (6), while in model R only $p_{A,B}^\omega$ is. Since for each model it is unambiguous which, $p_{A,B}^\lambda$ or $p_{A,B}^\omega$, is referred to, we neglect the superscripts in the following.

To summarize, we model the system as a continuous time Markov process with cells of type $A$ and $B$ arranged on a square lattice of length $L$ (that is, with $N = L^2$ lattice sites), and the possible transitions and parameters as in (3) and (4), together with the functional forms for $p_{A,B}$, (5) and (6), respectively. In particular, we study the model versions C and R, by fixing parameter values according to (7) and (8), respectively.

## Methods

To study the stochastic model numerically, we undertake computer simulations following a variant of the Gillespie algorithm [46], also called *random sequential update* [47]: during each *Monte Carlo step (MCS)*, associated with a time period defined by the total event rate as $\tau = \frac{1}{\lambda + \omega}$, we choose $N = L^2$ times a lattice site $i$ and one of its neighbours $j$, each randomly and with equal probability, and update site $i$ according to rules (3)–(6) (see discussion of the algorithm in the supplemental text of [48]). Update outcomes are according to the rules defined in the "Model" section, whereby in general any event that is possible (if the configuration allows it, as in (3)) and occurs with a rate, let's say, $\gamma$ (e.g. $\gamma = \lambda p_A^\lambda$ in the case of (3), left), is chosen

with probability $\frac{\gamma}{\omega+\lambda}$. Through repeated updates, the system evolves. The initial condition is a random distribution of cell types, with each cell type chosen with equal probability for each site. We generally choose a time long enough for the system to settle into a steady state before recording outputs (runtimes of $L^2/2$ MCS or more), except for the situation $\omega = 0$, $J = 0$, when the system is equivalent to the *voter model*, a model where sites randomly copy their state to a neighbouring site, without any further interaction [41]. This model has an equilibration time that diverges with increasing system size [34].

## Results

### Simulation results

We will now study the two model versions, C and R, numerically and will determine whether long-range order, such as large patches or other patterning, emerges. For convenience, we assign each lattice site $i$ a value $c_i = +1$ if it is occupied by a cell of type $A$, and we assign $c_i = -1$ if it is occupied by a cell of type $B$. This allows us to express $n_i = \sum_{j \sim i} c_j$ where $j \sim i$ denotes all sites $j$ neighbouring site $i$. To assess whether macroscopic patches of cells of equal type emerge, that are of comparable size as the whole epithelial sheet, we measure as an *order parameter* the difference in proportions of $A$ and $B$ cells, $\phi = |\frac{N_A - N_B}{N_A + N_B}|$, where $N_{A,B}$ are the total number of cells of types $A$, $B$ on the lattice. The order parameter is a widely used measure to identify phase transitions in complex systems [49, 50] We can also express this as $\phi = \frac{|\sum_i c_i|}{L^2}$, where the sum is over the whole lattice. The rationale of choosing this measure is that if patches are only small compared to the system size, and we let the system size $L$ be large ($L \to \infty$), then the proportions of $A$ and $B$ cells should become equal in this limit, and $\phi \approx 0$. However, if patches emerge that span a substantial fraction of the whole system, then one or few clusters of one type, $A$ or $B$, may dominate, leading to a non-zero value of the order parameter, $\phi > 0$. Similarly, we will assess a "staggered" order parameter $\tilde{\phi}$ [51], which measures the emergence of macroscopic patches of a checkerboard pattern, that is, alternating cell types. For that, we generate a 'staggered' lattice with site values $\tilde{c}_i = (-1)^{k_i + l_i} c_i$, where $k_i$, $l_i$ are row and column index of site $i$, respectively, and define $\tilde{\phi} = \frac{|\sum_i \tilde{c}_i|}{L^2}$. Thus, $\tilde{\phi}$ is effectively the order parameter $\phi$ taken of the staggered lattice. Since the values $\tilde{c}_i$ are generated by flipping cell types in a checkerboard pattern, any checkerboard pattern in $c_i$ becomes a patch of equal types in $\tilde{c}_i$. Therefore, $\tilde{\phi}$ measures the emergence of macroscopic patches of checkerboard patterns of cell types.

We simulated the model versions, C and R, for varying values of the interaction strength, $J$, and the proportion of symmetric divisions, $q = \frac{\lambda}{\lambda + \omega}$, and computed the order parameters $\phi$ and $\tilde{\phi}$. For model C, the results are displayed in Fig 2, both for a logistic cell-cell interaction function $p_{A,B}(n_i)$, according to (5) (left column), and the Hill function, (6) (right column). Notably, both these cases show the same behaviour: the order parameter $\phi$ is close to zero for any negative value of $J$, while it raises rapidly to substantially non-zero values for any $J \geq 0$. $\tilde{\phi}$, on the other hand, is close to zero for any value of $J$. Fig 2 also shows the distribution of cell types on the lattice (bottom), for a negative, positive, and zero value of $J$, with black pixels representing $A$-cells and white pixels representing $B$-cells. As suggested by the order parameters, for negative $J$ no ordering of cells is apparent, one neither sees large patches, nor patterns. For positive $J$, on the other hand, one sees large patches emerging, even filling the whole lattice. For $J = 0$ we also see large clusters, yet they look qualitatively different to the ones for $J > 0$: The clusters at $J = 0$ have very fuzzy borders, while those for $J > 0$ have more clearly defined patch borders. Hence, we can conclude that if cell fate is irreversibly chosen at cell division, the default behaviour, for no signalling interaction and for lateral induction, is that macroscopic

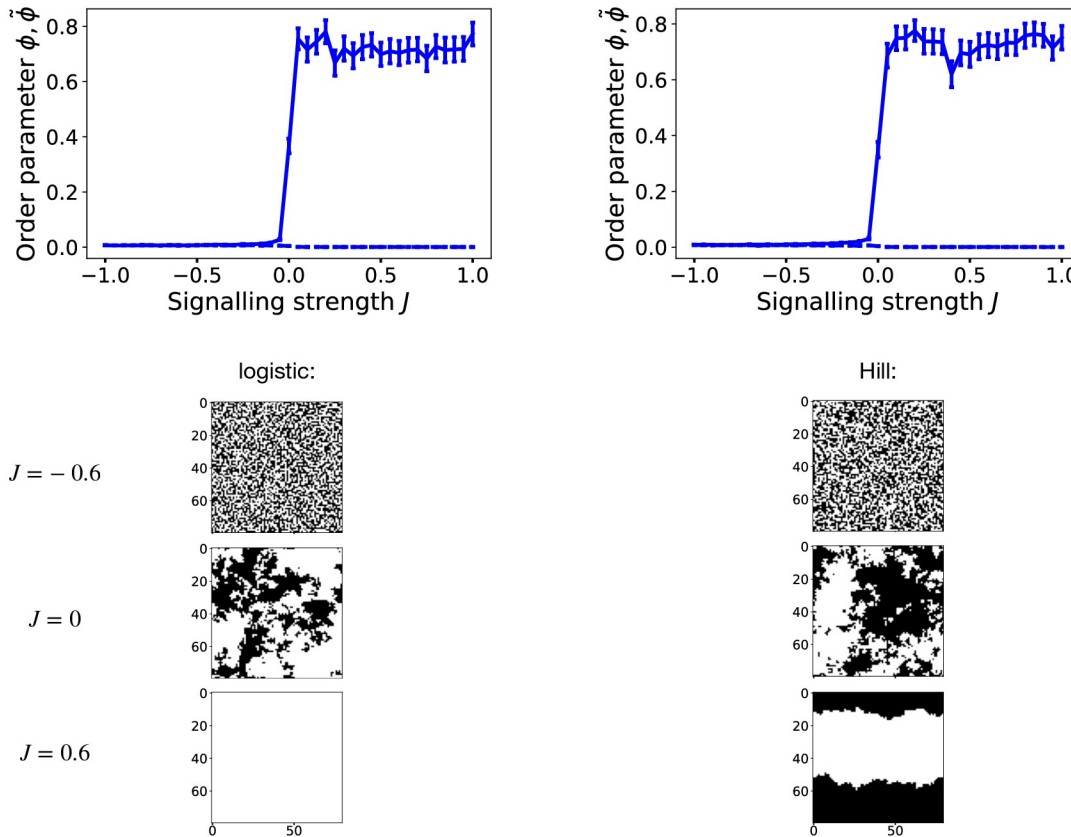

**Fig 2.** Simulation results for model C, **Left**: for a logistic interaction function, $p_{A,B}(n_i)$, according to (5), **Right**: for a Hill-type interaction function, according to (6). **Top row**: Order parameters $\phi$ (solid curve) and $\tilde{\phi}$ (dashed curve) as function of the signalling strength $J$. The curve shows the mean order parameters $\phi$ and $\tilde{\phi}$ of 80 simulation runs with the same parameters, and random initial conditions as described in the Methods section. Error bars are standard error of mean. The used lattice length is $L = 80$ ($N = L^2 = 6400$ sites) and we simulated for 4000 MCS until computing the order parameters. Below these are corresponding configurations of cell types on the lattice (black are A-cells, white are B-cells, and the tick labels denote lattice position), for logistic interaction function (**left**) and Hill-type interaction function (**right**), for different values of $J$ in each row.

cell type clusters, in size similar to the system size, emerge. Only lateral inhibition disrupts this order.

For model R, there are two parameters: $J$ and the proportion $q \coloneqq \frac{\lambda}{\lambda+\gamma}$ of symmetric cell division events. For $q = 1$ the model is identical to model C with $J = 0$, while for $q = 0$ there are no symmetric divisions and cell types switch reversibly with rate $\omega$ and probabilities $p_{A,B}$. Fig 3 shows the order parameters, both $\phi$ and $\tilde{\phi}$, for $p_{A,B}(n_i)$ being a logistic cell-cell interaction function according to (5) (left column), and a Hill function, according to (6) (right column). In contrast to model C, we see that for small magnitudes of $J$, both in negative and positive ranges, both $\phi$ and $\tilde{\phi}$ are close to zero and thus no long-range ordering emerges. However, at some critical point $J = J_c > 0$ the order parameter $\phi$ suddenly increases to substantially non-zero values. This feature occurs for both logistic and Hill-type interaction function, although $J_c$ is larger in case of signalling interactions following a Hill function. Furthermore, for $q = 0$ we see a transition in $\tilde{\phi}$ from zero to non-zero values if $J < \tilde{J}_c$ for some $\tilde{J}_c < 0$. We also see this when observing the configurations of cell types on the lattice (bottom of Fig 3): for sufficiently small values $J < \tilde{J}_c < 0$ and $q = 0$, large checkerboard patterns emerge, while for negative

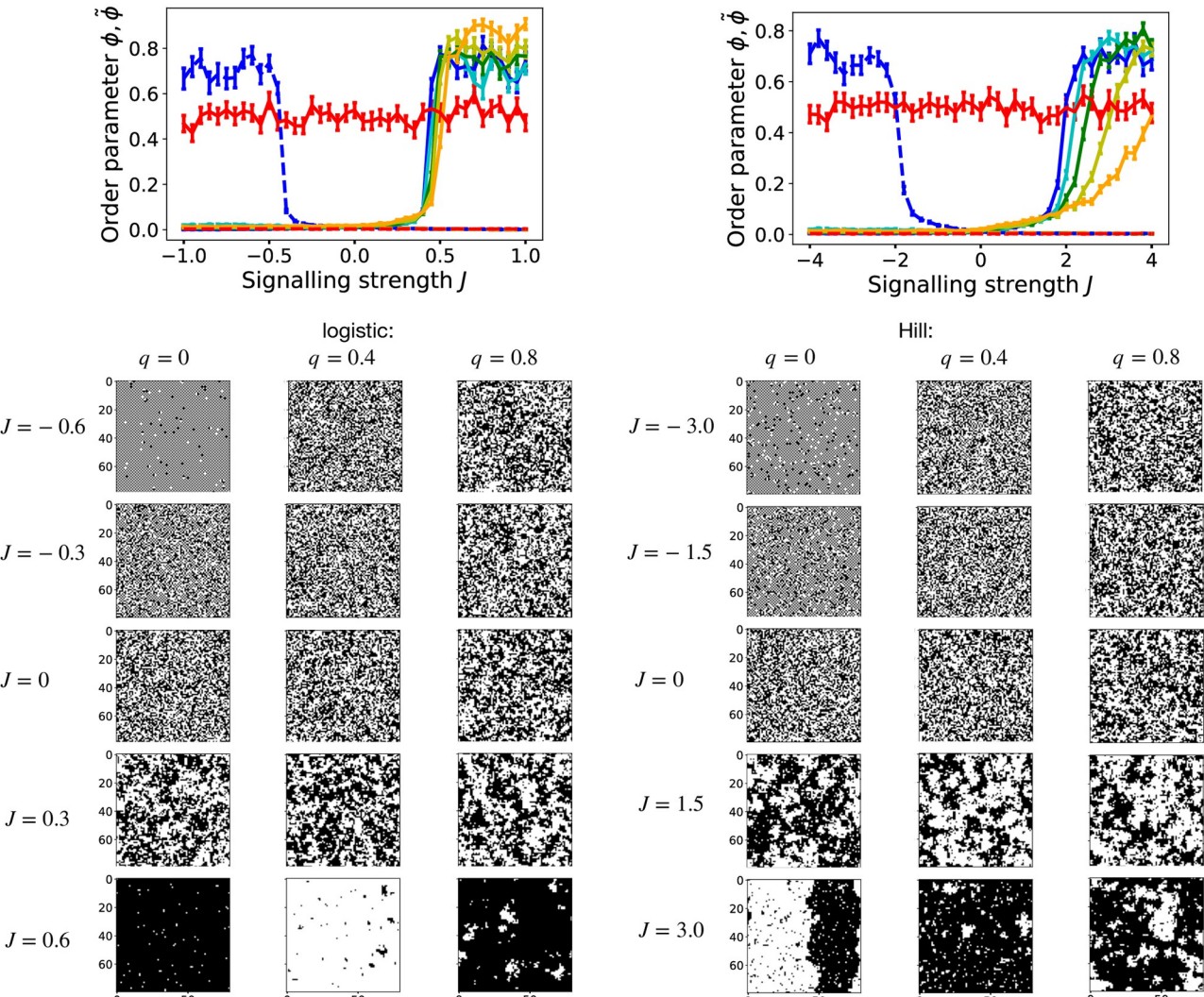

**Fig 3.** Simulation results for model R, **Left**: for a logistic interaction function, $p_{A,B}(n_i)$, according to (5), **Right**: for a Hill-type interaction function, according to (6). **Top row**: Order parameters $\phi$ (solid curves) and $\tilde{\phi}$ (dashed curves) as function of the signalling strength $J$, for model R, for different values of $q = \frac{\lambda}{\lambda + \omega}$: $q = 0$ (blue), $q = 0.2$ (cyan), $q = 0.4$ (green), $q = 0.6$ (yellow), $q = 0.8$ (orange), $q = 1$ (red). Each curve shows the mean order parameters $\phi$ and $\tilde{\phi}$ of 80 simulation runs with the same parameters, and random initial conditions as described in the Methods section. Error bars are standard error of mean. The used lattice length is $L = 80$ ($N = L^2 = 6400$ sites) and we simulated for 4000 MCS until computing the order parameters. Below these are corresponding configurations of cell types shown (black are A-cells, white are B-cells, and the tick labels denote lattice position), for different values of $J$ (rows) and $q$ (columns) as noted at the margins (note that values of $J$ differ between left and right panel arrays). Configurations for $q = 0$ and $J = -0.6$ (left) and $J = -3.0$ (right) display checkerboard patterns, which are seen best when zoomed in.

values of $J$ of less magnitude, $\tilde{J}_c < J < 0$, no long-range order is apparent, as is for small positive values $0 < J < J_c$. For large values of $J > J_c$ irregular large-scale patches emerge, for any value of $q > 0$. $J_c$ and $\tilde{J}_c$ differ between the logistic and Hill-type interaction function, but the qualitative features are the same. We can thus conclude that if cell fate is reversible, then a non-zero threshold interaction strength $|J|$ must be exceeded for long range order to emerge (macroscopic patches for lateral induction, alternating patterns for lateral inhibition). However, in contrast to model C, cell type patches and checkerboard patterns contain some defects, with some cells not matching the surrounding order, which is due to the non-zero probability to switch cell type even for cells deep in the bulk of a patch/pattern.

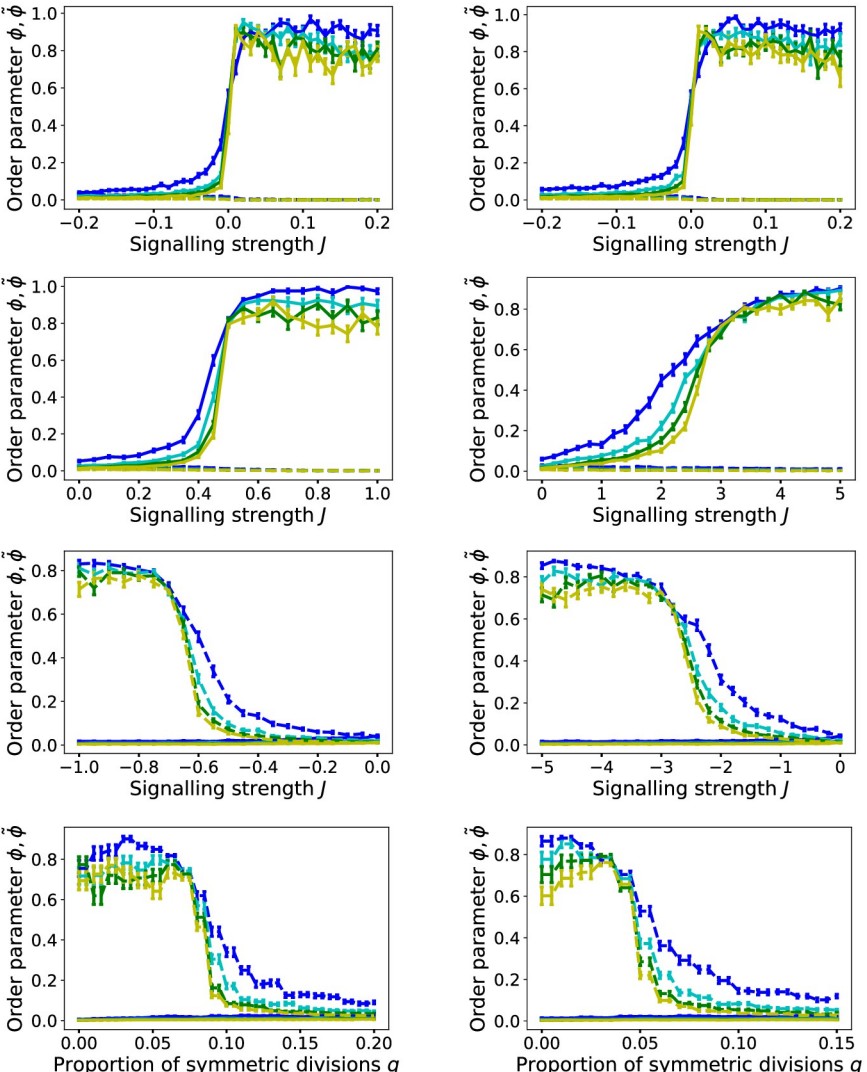

**Fig 4. System size scaling.** Simulated order parameters $\phi$ (solid curves) and $\tilde{\phi}$ (dashed curves) as function of $J$ and $q$, for increasing system sizes $L = 20$ (blue), $L = 40$ (cyan), $L = 60$ (green), $L = 80$ (yellow) and runtimes $L^2/2$ MCS. Each curve shows the mean order parameters $\phi$ and $\tilde{\phi}$ of 80 simulation runs with the same parameters, and random initial conditions as described in the Methods section. Error bars are standard error of mean. **Left column**: For logistic interaction function, $p_{A,B}(n_i)$, according to (5). **Right column**: For Hill-type interaction function, according to (6). **Top row**: $\phi(J)$ and $\tilde{\phi}(J)$ for model C. **2nd row**: $\phi(J)$ and $\tilde{\phi}(J)$ for model R, with $q = 0.5$. **3rd row**: $\phi(J)$ and $\tilde{\phi}(J)$ for model R, with $q = 0.05$ (**left**) and $q = 0.02$ (**right**). **Bottom row**: $\phi(q)$ and $\tilde{\phi}(q)$ for $J = -2.0$ (**left**), and $J = -5.0$ (**right**).

We now wish to test whether the observed transitions from $\phi, \tilde{\phi} \approx 0$ to substantial non-zero values are genuine phase transitions, that is, a non-analytic transition from strictly $\phi = 0, \tilde{\phi} = 0$ to non-zero-values at $J_c$ and $\tilde{J}_c$, when $L \to \infty$. Phase transitions are strictly only defined in infinitely large systems, but here we are limited by computational constraints to finite systems. Yet, we can assess this problem by scaling the system size. We show the results in Fig 4. Here we see that the transitions from $\phi, \tilde{\phi} \approx 0$ to $\phi, \tilde{\phi} > 0$ become indeed sharper with increasing system size in either model, indicating that $\phi, \tilde{\phi} \to 0$ for $L \to \infty$ in the regime

$\tilde{J}_c < J < J_c$, as required for a phase transition. Intriguingly, we see the transition from $\tilde{\phi} = 0$ to $\tilde{\phi} > 0$ in model R also for non-zero but small values $q > 0$ (Fig 4, 3rd row). Furthermore, if we vary $q$ for sufficiently small $J < \tilde{J}_c$, we see that the non-zero regime of $\tilde{\phi}$ prevails also for non-zero values of $q$ as long as $q < q_c$ for some critical threshold value $q_c$, beyond which it drops sharply to zero (Fig 4, bottom row). Also for this transition, the profile become sharper with system size. This indicates that the ordered phase with macroscopic checkerboard patterns prevails for small, but non-zero proportions of symmetric divisions, $q$, and only for $q > q_c$ long-range order vanishes. Again this qualitative behaviour is seen for both the logistic and Hill-type interaction function, only the numerical values of $q_c$ vary.

## Theoretical insights

To understand the observations made by simulations, we can get insights by mapping the model on a generic two-state spin system as employed in statistical physics. As stated before, we interpret the cell types as numbers $c_i = \pm 1$ which in spin systems can be interpreted as "spin up" ($\uparrow$,+1) and "spin down" ($\downarrow$, -1). We now consider a particular class of spin systems, which we here call *memoryless spin-update models (MSUM)*, as studied in [52]. Such systems are defined by (1) individual sites being randomly chosen, with equal probability, and updated, (2) the probabilities that after the update a spin has value $\pm 1$, called $p_\pm$, may depend on the neighbours of site $i$, but not on the value of the spin $c_i$, itself, before the update (hence $p_\mp = 1 - p_\pm$), (3) the spin update probability is symmetric with respect to the neighbour configurations, that is, $p_\pm(-n_i) = 1 - p_\pm(n_i)$. Such systems have been studied and well understood by means of statistical physics [52]. This class of models contains both the voter and the Ising model [34] for particular parameter values. Notably, due to the symmetries of $p_\pm(n_i)$, these functions, and thus the model as a whole, are completely defined by two parameters, namely,

$$p_1 := p_+(2), p_2 := p_+(4) \ , \tag{9}$$

since due to the symmetry of the function $p_+(n_i)$, all other possible values of $p_+$ are fixed as $p_+(0) = \frac{1}{2}, p_+(-2) = 1 - p(2), p_+(-4) = 1 - p(4)$, and further $p_- = 1 - p_+$ is fixed (odd values and values outside the range $[-4, 4]$ are not possible, as only the four neighbours of $i$ are considered). In ref. [52] it has been shown that such a system displays a phase transition in the $p_1$-$p_2$ parameter plane between an ordered and a disordered phase. This phase transition is of the same universality class as that of the Ising model, except for the particular point $(p_1, p_2) = (3/4, 1)$ at which the system corresponds to the voter model. There, any cluster, which in contrast to the Ising model class have fractal surfaces, diverges over time, so that $\phi \to 1$ for any finite system, yet the mean equilibration time is infinite. A sketch of the phase diagram in the $p_1$-$p_2$-plane is shown in Fig 5, where the dashed black curve sketches the phase transition line.

Now we assess whether our model can be interpreted as a MSUM. First, we consider the rates at which a single site on the lattice is updated according to the model rules (3) and (4). Without loss of generality, let us consider a particular site $i$ on the lattice that contains a $B$-cell. The rate for this site to change its occupation to an $A$-cell, either by a change of the cell's identity or by being replaced by an $A$-cell via the symmetric division of a neighbour, is composed of the rates of two events: (1) the cell type changes according to events (4), with rate $\omega p_A^\omega$, or (2) according to events (3) an event $(A, B) \to (A, A)$, occuring with rate $\lambda p_A^\lambda$, turns a $B$ cell into an $A$ cell. However, this may occur both if the $B$ cell on site $i$ is selected and if the neighbouring $A$ cell is selected, thus the total rate for this to occur is doubled, $2\lambda p_A^\lambda$. Hence the total rate at

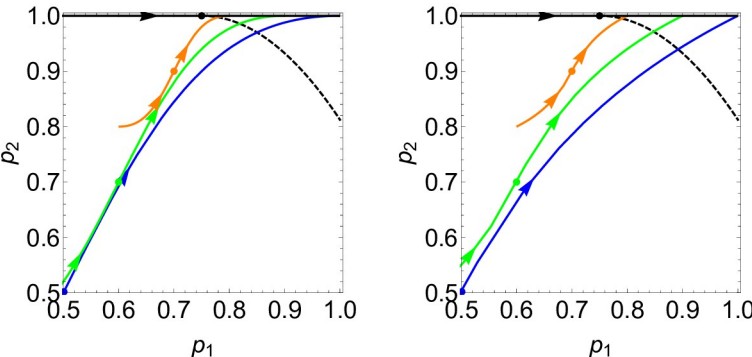

**Fig 5. MSUM $p_1$-$p_2$ phase space.** Depiction of our model's implied MSUM parameters $p_1$ and $p_2$ as function of $J$, $\boldsymbol{p}(J)$ = $(p_1(J), p_2(J))$, within the $p_1$-$p_2$ parameter plane, according to (13) and (15) (when substituting (5) and (6), respectively). Displayed are curves for model C in steady state (black) and for model R and different values of $q$: $q = 0$ (blue), $q = 0.4$ (green), $q = 0.8$ (red). **Left**: for a logistic interaction function, (5). **Right**: for a Hill-type interaction function, (6). The dots on curves denote the $(p_1, p_2)$ values for $J = 0$, and the arrows show the direction of increasing $J$. The dashed black line is a sketch of the phase transition line according to [52], which is of the Ising universality class, except for the point $\boldsymbol{p}_v = (0.75, 1)$ which corresponds to the voter model (no exact form for the phase transition curve is available, except for the point $\boldsymbol{p} = \boldsymbol{p}_v$).

which a $B$ cell on site $i$ becomes/is replaced by an $A$-cell is,

$$\gamma_A(n_i) = \omega p_A^\omega + f_i^A 2\lambda p_A^\lambda = \frac{n_i + 4}{8} 2\lambda p_A^\lambda + \omega p_A^\omega , \qquad (10)$$

where $f_i^A = \frac{n_i+4}{8}$ is the probability that a randomly chosen neighbour of site $i$ is of type $A$, so that an update according to (3) can occur. For an $A$ cell, the rate to change cell type is analogously,

$$\gamma_B(n_i) = \gamma_A(-n_i) = \frac{-n_i + 4}{8} 2\lambda p_B^\lambda + \omega p_B^\omega . \qquad (11)$$

To see whether this continuous time stochastic process is equivalent to a MSUM, we analyse the random-sequential update scheme (Gillespie algorithm) we used for simulating the system (see "Methods" subsection). We start with model R, that is, setting $p_A^\lambda = p_B^\lambda = \frac{1}{2}$. If we choose as time unit the update scheme's Monte Carlo time steps $\tau = \frac{1}{\lambda+\omega}$, the probability that a randomly selected site $i$ with a $B$-cell becomes an $A$-cell after a Monte Carlo update is $p_{B\to A} = \gamma_A \tau = q \frac{n_i+4}{8} + (1-q)p_A(n_i)$. Similarly, we get the probability for an $A$-cell to become a $B$-cell, $p_{A\to B} = \gamma_B \tau = q \frac{-n_i+4}{8} + (1-q)p_B(n_i)$. Crucially, the probability for an $A$-cell to stay an $A$-cell is $p_{A\to A} = 1 - p_{A\to B} = q \frac{n_i+4}{8} + (1-q)p_A(n_i) = p_{B\to A}$. Hence, the probability that after the update, site $i$ is occupied with an $A$-cell is $p_+ := p_{B\to A} = p_{A\to A}$, i.e.,

$$p_+^{(2)}(n_i) = \frac{n_i + 4}{8} q + (1-q)p_A(n_i) , \qquad (12)$$

where $p_+^{(2)}$ is independent of the occupation of $i$ before the update, whether $A$-, or $B$-cell (the superscript indicates the model version). The same is valid for $p_- = p_{A\to B} = p_{B\to B} = 1 - p_+$. Furthermore, the function $p_+(n_i)$ is symmetric with respect to the sign of $n_i$, $p_+(-n_i) = 1 - p_+(n_i)$

and thus model R is equivalent to an MSUM with the relevant parameters, according to [52],

$$
\begin{aligned}
p_1^{(2)} &= p_+^{(2)}(n_i = 2) = \frac{3}{4}q + (1-q)p_A(2) \\
p_2^{(2)} &= p_+^{(2)}(n_i = 4) = q + (1-q)p_A(4) \;,
\end{aligned}
\tag{13}
$$

where $p_A$ can take the two forms of interaction functions according to (5) and (6). We further note that $p_A = p_A(n_i, J)$ is also a function of $J$ and thus $p_1 = p_1(J, q)$ and $p_2 = p_2(J, q)$ are functions of both $J$ and $q$. In Fig 5, we show trajectories $\boldsymbol{p}(J) = (p_1(J), p_2(J))$ in the $p_1$-$p_2$ parameter plane for several values of $q$ (coloured curves), compared to a sketch of the Ising-type phase transition line of MSUMs [52] (black dashed line). We note that those trajectories cross the theoretical Ising phase transition line for values $J_c > 0$, for any $q < 1$. This confirms that model R indeed exhibits a phase transition of the Ising universality class at non-zero $J_c > 0$, that is, we see a "ferromagnetic" phase transition from a disordered phase, with order parameter $\phi = 0$ to an ordered phase with $\phi > 0$ that exhibits patches of cell types (i.e. spins) of a size comparable to the system size. The exception is $q = 1$, when the model is identical to the voter model (see discussion of this case below).

We note that switching to the staggered lattice, $c_i \to \tilde{c}_i$, corresponds to replacing $n_i \to -n_i$, since either only $c_i$ flips sign or all its neighbours. For $q = 0$, we have $p_\pm = p_{A,B}$ and since $p_{A,B}$ are functions of $Jn_i$, $p_\pm$ are symmetric towards the transformation $c_i \to \tilde{c}_i, J \to -J$. Hence, it is expected that $\tilde{\phi} = \phi(\{\tilde{c}_i\})$ exhibits the same phase transition at $\tilde{J}_c = -J_c$ as $\phi$ does at $J_c$, yet via emergence of checkerboard patterns instead of patches of equal cell types. This is consistent with the phase transition we observed numerically for $q = 0$ and confirms that $\tilde{J}_c = -J_c$. However, we also observe numerically a phase transition for small non-zero values $q > 0$, in which case the system is not symmetric with respect to $J \to -J, \phi \to \tilde{\phi}$. To understand this, let us consider a situation when $q > 0$ is very small, and $J < \tilde{J}_c$, i.e. when $\tilde{\phi} > 0$. This corresponds to the situation where $J > J_c$ and $\phi > 0$ on the staggered lattice of spins $\tilde{c}_i$, i.e. when the system is within the ordered region of the $p_1$-$p_2$ phase diagram (upper right corner in Fig 5). Any symmetric division within a checkerboard patterned area flips the cell type at one site $i$, leading to $\tilde{c}_i \to -\tilde{c}_i$. On the staggered lattice, this corresponds either to a transition $A \to B$ when $\tilde{n}_i = 4$ or $B \to A$ when $\tilde{n}_i = -4$, meaning that effectively, the probability of symmetric divisions, $q$, lowers the probability $p_2$, that is $p_2 \to p_2 - q$. This corresponds to a shift in the parameter plane as $(p_1, p_2) \to (p_1, p_2 - q)$. If $q$ is small enough, the system remains within the regime of the ordered phase (beyond the black line in Fig 5), while if $q$ becomes larger, it may cross the Ising phase transition line towards the disordered phase.

For model C, we cannot find a symmetric update probability in general, for any fixed time unit $\tau$. However, if we assume the system to be in the steady state, we can devise an update algorithm that corresponds to an MSUM: as the steady state is time-invariant, we can choose the time unit between updates individually for each update, and do not need to define an absolute time unit. Thus, as before, we undertake a random-sequential update scheme, selecting sites randomly, but use at each update of site $i$ a different time interval between updates, namely $\tau_i = \frac{1}{\gamma_A(n_i) + \gamma_B(n_i)}$. We also simplify the interaction by assuming that the probabilities of updates of site $i$ depend only on the neighbouring sites of site $i$, and not on those of the other site $j$ involved in a cell division according to (3). Since $i$ and $j$ will be chosen at equal probabilities over time, the joint update probabilities of sites $i$ and $j$ depend on all 6 neighbours of $i$ and $j$, as in our numerical model, and thus in the time-invariant stationary state, the model outcomes of this MSUM are expected to be equivalent to our numerical model from previous sections. Then we get $p_{B \to A} = \frac{\gamma_A(n_i)}{\gamma_A(n_i) + \gamma_B(n_i)} = \frac{(4+n_i)p_A}{(4+n_i)p_A(n_i) + (4-n_i)p_A(-n_i)}$, where

we used that $p_B(n_i) = p_A(-n_i)$. Furthermore, $p_{A \to A} := 1 - p_{A \to B} = 1 - \frac{\gamma_B(n_i)}{\gamma_B(n_i) + \gamma_A(n_i)} = p_{B \to A}$, thus the update outcome is independent of the initial value on site $i$. This means that in the steady state we can define a probability to update to an $A$-cell, $p_+^{(1)}(n_i)$, being independent of the value on site $i$, as required for a MSUM:

$$p_+^{(1)} = \frac{(n_i + 4)p_A(n_i)}{(n_i + 4)p_A(n_i) + (4 - n_i)p_A(-n_i)} \ . \tag{14}$$

This update probability is also symmetric, $p_- = 1 - p_+$ and $p_+(-n_i) = 1 - p_+(n_i)$. Hence, model C in the steady state, with the approximation to count only neighbours of the updated site $i$, constitutes a MSUM. The corresponding relevant parameters are,

$$
\begin{aligned}
p_1^{(1)} &= p_+^{(1)}(n_i = 2) = \frac{3p_A(2)}{1 + 2p_A(2)} \\
p_2^{(1)} &= p_+^{(1)}(n_i = 4) = 1 \ .
\end{aligned}
\tag{15}
$$

Again, we see the trajectory $\boldsymbol{p}(J)$ plotted in Fig 5 (black line), which is on the top edge of the diagram, at $p_2 = 1$. Notably, the trajectories for the different interaction functions as given in (5) and (6) both show the same key features: for $J = 0$, we have $p_A(2) = 1/2$ and thus $p_1 = 3/4$, which is exactly the critical point corresponding to the voter model. For any negative $J$, the system is in the disordered regime, left of the transition line, while for any positive $J$, it is in the ordered regime, right of the line. Hence, the transition from disordered, with $\phi = 0$, to ordered, $\phi > 0$, occurs exactly at $J = 0$, as we have observed numerically. However, the phase transition is of a different character than the Ising model phase transition. In fact, at the critical point, for $J = 0$, the system is equivalent to the voter model, which does not exhibit a steady state for any infinite system with $L \to \infty$. For any finite system, it will eventually lead to $\phi = 1$, with one species, $A$ or $B$, occupying every lattice site; however, the expected time for this to occur is infinite.

## Discussion

We analysed a cellular automaton model for the cell population dynamics in an epithelial sheet, by modelling cells of two possible types, a stem cell type ($A$), which can divide, and a cell type primed for differentiation ($B$), which does not divide, set in a square lattice arrangement. We modelled cell division and fate dynamics according to established models of cell fate choice [19, 21, 40], but assumed in addition that fate choices are regulated by juxtacrine signalling between neighbouring cells. These dynamics mimic, for example, cells in the basal layers of epidermis [21], oesophagus [53], or organotypic cultures [54], which are smooth sheets or have tubular geometry, and which may be regulated through juxtacrine Notch-Delta or Notch-Jagged signalling. We assessed the spatial distributions of cell types in the lattice, as generated from two biologically motivated versions of the model: in one version we assumed that a cell commits to its fate when it divides, while in the other version, changes of cell type can occur independently of cell division, in a reversible manner that reflects 'licensing' to differentiate [22]. In either case, we assumed that the propensity of cell fate choice is regulated through signalling which is either "laterally inducing", preferring the choice of the cell type as the majority of neighbouring cells, or "laterally inhibiting", preferring the opposite cell type to that of the majority of neighbours. We modelled this interaction through a probability of fate choice that depends on the number of neighbours of either cell type, through two possible functional forms, a logistic and a Hill-type function. The strength of this interaction is

quantified by a single parameter $J$, whereby positive $J$ corresponds to a laterally inducing interaction, and negative $J$ corresponds to a laterally inhibiting interaction.

Through numerical simulations that we confirmed by theoretical considerations, we found that when cell fate is committed and coupled to cell division, the system usually exhibits long-range order, where macroscopic homogeneous patches (cells of equal type) of size similar to the system size emerge whenever there is no regulating interaction or it is laterally inducing. Only for laterally inhibiting interaction, no long-range order is observed. If cell fate is reversible and is regulated independently of cell division, long-range order is generally only observed if the interaction strength $|J|$ exceeds a critical threshold value $J_c > 0$. If signalling is laterally inducing and is sufficiently strong ($J > J_c$), macroscopic homogeneous patches emerge. If the proportion of symmetric divisions is sufficiently low, long-range order emerges also for sufficiently strong laterally inhibiting interactions, if $J < -J_c < 0$, in which case large-scale patterns of alternating cell types, arranged like a checkerboard, emerge. For $|J| < J_c$, no long-range order is observed. The observed features are independent of the functional form chosen to model the signalling interaction between cells, both a logistic function and a Hill-type function show the same qualitative behaviour. This means that for modelling such qualitative features, one can choose the type of interaction function freely; preferably such that the analysis is simplified accordingly.

The association of patterns and cell type patches with juxtacrine signalling pathways has been demonstrated previously in various works: lateral inhibition can lead to alternating cell type patterns [10, 30, 32] and lateral induction to patches of cells of the same type [3, 14, 55, 56]). Our work shows that also cell division and associated cell fate choice dynamics are crucial factors to account for when assessing such large-scale features of cell type arrangements. For example, the emergence of alternating patterns, under lateral inhibition in our model, is only possible if cell fate is reversible *and* if divisions are predominantly asymmetric. This means that symmetric cell divisions generally suppress alternating cell type patterns. On the other hand, large-scale patches can emerge from lateral induction *or* from symmetric divisions when cells commit to differentiation, even in absence of any regulation.

Our model, like any cellular automaton model, is subject to simplifications that may lead to deviations of quantitative predictions compared to the real world situation. For example, our model has a fixed fourfold rotational symmetry, the cell arrangement is fixed, and mobility is only possible through replacement of lost cells. Despite these simplifications, we expect qualitative features of emergent phenomena to prevail in reality, such as the occurrence of a phase transition at some critical point of parameter values. This is a consequence of 'universality' [43, 44], the phenomenon that often only few model features such as symmetries, dimension, and conserved quantities are relevant for qualitative features, while model details do not affect those. Other features may only partially prevail: for example, it is unlikely that genuine checkerboard patterns emerge in reality, as these are a feature of the square lattice's fourfold symmetry, but approximately alternating patterns would generally be expected (For hexagonal cell arrangements ('triangular' lattices), which resemble real-world cell arrangements more closely, no exact alternating pattern is possible [57], but one that is close to alternating, with some defects interspersed). Beyond this, certain assumptions of our model are possibly more accurate than expected: in mouse epidermis, it was shown that cell arrangements do not change much over time and that cell loss is accompanied by direct replacement through division of a neighbouring cell [24]. Yet, our model can only be the starting point and theoretical groundwork for a future comprehensive modelling framework which will need to explore more detailed models and test quantitative features on experimental data, for example by including cell intercalation when implemented as a vertex model [27, 58, 59]. Finally, our model is only able to test—and thus possibly exclude—hypotheses within its scope, that is, with juxtacrine

nearest-neighbour signalling interaction. Long-range signalling through diffusive ligands or long membrane protrusions [38, 39] are not considered here and can only be tested by models which explicitly include those signalling mechanisms.

The question whether cell fate is being decided at cell division or independently of it is a long-standing one and has only recently been decided experimentally in a few tissues [24–26], through rather complicated and expensive intra-vital imaging assays. Hence, experimental approaches which are feasible and not too expensive are desirable, as the commonly used method of (static) genetic cell lineage tracing combined with clonal modelling turns out to be insufficient to distinguish these cases [19, 20]. The close association of cell fate choice and large-scale features of the cell type arrangement suggests that experiments which can measure this arrangement could be used, in conjunction with mathematical modelling (using our model or future more detailed models), to answer those questions. A candidate approach to measure this are 3D confocal immunofluorescent assays, employed to obtain images of tissues with molecular markers that identify cell types relevant for cell fate choices and regulation. Such experiments have been done extensively in many tissues, but a comparison with the models is not straightforward without further advanced image processing, as the experimental data does not necessarily reflect the 2D arrangement of cells in epithelial sheets that are not entirely flat. As such, the 3D immunofluorescent images first need to be 'unfolded' into a 2D arrangement of cells through image analysis and topological algorithms that preserve cell-cell contacts, and to analyse them. Following this, the order parameter or other measures, such as the correlation function or topological methods (e.g. persistent homology [60]) that can identify further features of the cell type arrangement, can be be used to test the models.

The so processed experimental data can then be used to test, and possibly reject, certain hypotheses on cell fate choice. For example, assume we knew that laterally inducing juxtacrine signalling is prevalent, then the absence of long range ordering suggests that cell fate is reversible, as we have seen that only model R may lack order, for sufficiently small interaction strength. The observation of alternating cell type patterns, such as in chick inner ear (see Fig 1B), also requires that cell fate choice is reversible, and furthermore, it implies that the proportion of symmetric divisions must be rather small.

To summarise, this work shows that qualitative features of spatial cell type arrangements, such as long-range order, express information about the underlying modes of cell fate choice. By analysing those features experimentally, conclusions about the reversibility of cell fate, and whether cell fate is decided at cell division or independently of it, can be drawn.

## Acknowledgments

The author thanks Ceres Gijsels and Yoshiki Cook for preliminary work related to this article's subject.

## Author Contributions

**Writing – original draft:** Philip Greulich.

**Writing – review & editing:** Philip Greulich.

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
