## [Decision Letter · Decision Letter 0]

19 Oct 2023

Dear Dr Greulich,

Thank you very much for submitting your manuscript "Emergent order in epithelial sheets by interplay of cell divisions and cell fate regulation" for consideration at PLOS Computational Biology.

As with all papers reviewed by the journal, your manuscript was reviewed by members of the editorial board and by several independent reviewers. In light of the reviews (below this email), we would like to invite the resubmission of a significantly-revised version that takes into account the reviewers' comments.

We cannot make any decision about publication until we have seen the revised manuscript and your response to the reviewers' comments. Your revised manuscript is also likely to be sent to reviewers for further evaluation.

Sincerely,

Philip K Maini

Academic Editor

PLOS Computational Biology

Jason Haugh

Section Editor

PLOS Computational Biology

Reviewer's Responses to Questions

**Comments to the Authors:**

Reviewer #1: The article “Emergent order in epithelial sheets by interplay of cell divisions and cell fate regulation” looks at how an agent based model of cell fate choice may be used to better understand emergent order of cell-fate selection, and the effects cell division of fate selection.

This work begins with a brief motivation of the problem of cell-fate selection, and then proceeds to discuss some previous attempts in a brief manner. The authors outline their simple, yet very elegant agent based model which describes cell fate selection and division. To decide how cells transition between difference cell types (in turn, how their fate is chosen), the authors propose two different limiting probability distributions, and compare the models sensitivity to them, overserving how spatial patterning arises in both cases. To very different mechanisms in cell-fate selection are described: a cells fate is determined upon division, or cells may change their fate dynamically. A qualitative analysis is provided, comparing the behaviour from the different approaches.

I have a few comments and concerns regarding the articles fit for the journal PLoS Computational Biology.

1: The article requires a more concise literature review. Ising models have been well studied in physics with many people using them to describe cell behaviour in biological tissues. Some more effort to distinguish why this is a suitable approach to understand the emergent order of cell-fate selection, and also presenting these previous approaches would make this a much stronger article in it’s current form.

2: Upon reading the article, the work should be related back to the biological system in a clearer way. There are some rather interesting results that are observed in the article, but they are not related to a biological system.

3: Within the (rather nice and appealing) simulation results shown, they are presented for only single instance results. I would like to see some analysis perform with various simulations and a statistical analysis perform, to ensure confidence in the numerical results.

4: I am concerned about this articles suitability for the journal PLoS Computational Biology. Typically, work requires a strong link to the biological system at hand. Relevant biological data needs to be incorporated. For example, if the article were to be considered for publication, I would suggest incorporating florescence imaging analysis to draw conclusion about which of the two proposed mechanisms best relate to the biology and how. Without the link, this work is much more of a theoretically valid approach.

Reviewer #2: In this manuscript, the author develops an on-lattice model of two cell types (A and B) undergoing division and differentiation according to specific rules. The stochastic model evolves according to a Gillespie algorithm, with sites on the lattice randomly selected and evaluated for differentiation or division. The author introduces two order parameters (phi and tilde{phi}) and performs simulations to identify how changes in the cell-interaction strength and proportion of symmetric divisions impact pattern formation. The simulations produce random distributions of cells, clumps of cells, and nearly complete separation of different types of cells. The author also relates their model to Ising-style models from statistical physics and discuses phase transitions.

This is an interesting paper and an enjoyable read. I am not convinced that it is appropriate for PLOS Computational Biology, however, since the model is not tightly tied to biological data. I also have questions about the order parameters and variability across stochastic simulations, and I would suggest adding more references to the previous literature on voter models, order parameters, and cell-based models in general.

Main comments:

(1) Have the order parameters that the author defines been used in other studies, and are there other order parameters or quantitative approaches that could be used, such as pair correlation functions or pattern simplicity scores that might provide more information and further validate the observed differences? In Figures 1-3, would it be possible to include the mean order parameter values across 100 simulations with standard deviation bars? Can these order parameters capture differences in cluster size or pattern width?

(2) At line 85 on page 5, I do not follow why phi = sum_i c_i is the same as phi = (N_A - N_B)/(N_A + N_B). Additionally, could the author add more intuition for what tilde{phi} is doing on an example pattern? Does tilde{phi} only capture checkerboard patterns with a pattern width of one grid square?

(3) At line 8 on page 2, I suggest including biological images from some papers showing the different patterns that the author is referring to, or a cartoon illustration that shows different types of regular patterns and irregular domains. Additionally, after Equation 3, could the author include references and more biological discussion about the assumption that stem cells maintain their position upon asymmetric division?

(4) For a general audience, it would be helpful to extend the discussion about how the model is similar to the voter model in the “Methods” section on page 5.

(5) In the section “Simulation results”, does c_i = 0 if it is empty, or is the number of cells always equal to the number of lattice sites?

(6) In several figures, legends and information about the meaning of different colors is missing from the caption. For example, in Figure 1, what do black and white refer to? In Figure 2, what is the gray color in the patterns, and what parameter values do the colors of the curves in the upper two figures refer to? In Figure 3, what color curves represent what parameter values? Additionally, what are the initial conditions for the figures?

(7) In the Discussion section, I would suggest weakening the language surrounding conclusions and adding some discussion of model limitations, since the author is considering a simplified model with several assumptions (i.e., related to the model being a square lattice, having two cell types, etc).

Minor comments:

(1) In Equations 5 and 6: what is J? I did not see this defined until later in the manuscript.

(2) At line 60 on page 5: I think an “and” is missing in the list after “To summarize…”

(3) On page 6, typo: “fuzzy borders”, not “fuzzy boarders”.

(4) On page 10, typo: “this corresponds” rather than “his corresponds”.

(5) On pages 10 and 11: I recommend moving more equations out of the paragraph form, so they appear on their own lines. This will make the paragraph more readable.

(6) My understanding is that the author refers to “macroscopic domains” as patterns with large, separated regions of A and B cells. To me, “domain” means the domain of the simulation. I wonder if there is a better name for these types of patterns?

(7) On page 8: The sentence “We now consider a particular class of spin systems,…” is too long and wordy and should be split into several sentences on page 8. The sentence “The rate for this site to change..” is also long and difficult to parse on page 8.

Reviewer #3: The analysis presented in the paper appears mostly correct (subject to requested clarifications below) and is mathematically appealing. However, the biological conclusions presented are not substantiated, and a wide range of existing literature in this area is not discussed. My recommendation is thus that the paper is reconsidered after revisions, or rejected.

In the paper, the author introduces a cellular automaton model that investigates the effect of asymmetric vs symmetric cell divisions on patterning through lateral inhibition or lateral induction. He documents phase transitions in his model, i.e. he identifies critical signalling strengths for patterning to occur or not occur, as measured by a global order parameter. The author shows that the model is equivalent to an Ising model of spin interactions, and thus identifies parallels between the phase transitions in both models. These observations are interesting and well-suited for an audience of mathematicians or physicists.

In the model, the extent of patterning differs between scenarios considering symmetric or asymmetric cell divisions, and scenarios in which cell fate commitment is separate from the division. The premise of the paper is that such differences in patterning can hence be used to distinguish between different models of cell fate commitment and division based on imaging data alone.

I have the following concerns regarding the premise of the paper:

1)The model makes a few biological assumptions that are quite strong: The simulated tissue is under homeostasis, i.e. cell proliferation is assumed to be balanced by epithelial cell delamination. Is there a biological tissue that fulfils this assumption and which also patterns via lateral inhibition or lateral induction? Maybe there are plenty systems like this and I am ignorant of them, but no such system is cited in the paper. The biological references that the author is citing seem to be mostly systems in which cells divide without delaminating, or in which cells are not reported to divide. Some of the tissues are actively growing.

Similarly, the model makes the strong assumption that if a cell divides, a neighbouring cell delaminates, i.e. cell delamination and cell division are locally coupled. Is this a true phenomenon that can be seen in epithelia? Even if we assume that the tissue is overall in homeostasis, wouldn’t we expect that the choice of delamination sites can affect the pattern just as much as the type of division events? If so, is it then justified that the model accounts for one of the effects but simplifies the other?

I believe naming biological systems to which the model applies is essential for the premise of the paper. If no system exists to which the modelling assumptions apply, then the point that the model can be used to infer differentiation processes from imaging data does not hold. In that case there remains merit to the presented theory, but I believe that theory should then not be presented under a false premise.

2)Lateral inhibition is biologically typically mediated by interactions between the Delta and Notch signalling pathways. Much is known about these interactions. There are more than 2 decades of research on this topic, including a wide range of literature investigating how the coupling of Delta and Notch can lead to different patterns. For example, Wearing et al (2000) showed how different patterning spacings may be achieved through incorporating positive feedback and Hadjivasiliou et al (2016) showed how long-range coupling can change the geometry of the pattern. Hawley et al (2022) investigated how patterns may change in time. – These are just a small selection of what’s out there. Since there are multiple factors affecting the existence and shape of geometric patterns, how can we use the patterns as a way to identify different modes of cell differentiation?

3)Is it true that we really don’t know the difference between symmetric and asymmetric cell divisions in the biological contexts of lateral inhibition and lateral induction? One reference that the author cites, a nature paper by Sprinzak et al from 2010, uses live imaging to investigate how Notch signalling changes over time. In their pictures (e.g. figure 3D) it looks like Notch signalling gradually increases over a time period of multiple cell divisions, suggesting that cell fate and cell divisions are uncoupled (or coupled in a way that cannot be represented by the author’s proposed model).

I have the following comments on the execution of the manuscript that I believe should also be addressed before acceptance.

4)The author refers to their simulation algorithm as a Gillespie algorithm while counting steps as ‘Monte Carlo steps’. Is it a Gillespie or a Monte-Carlo simulation? The algorithm seems different from a Gillespie algorithm to me, since at each update step lattice sites are picked with equal probability. At the same time, the propensity of an event occurring is described to vary between lattice sites, since a cell can’t divide if it doesn’t have at least one differentiated neighbour. It is my understanding that in a Gillespie algorithm, a lattice site would be chosen with a probability proportional to the propensity of an event occurring at the site, not with equal probability for each lattice site.

5)The author claims that the system reaches a steady state after L^2 lattice updates. Could this be demonstrated by showing the dynamics of the system over time, maybe in a supplementary figure? After L^2 lattice updates, each lattice site will on average have been updated once. My expectation would be that each lattice site would need to be updated multiple times before an equilibrium is reached.

6)In figures 1, 2, and 3, one to five repeats of a parameter variation are shown. While it’s nice to see the extent of stochastic variation by eye, I think it would be nicer to see means and error bars from a larger number of simulations. Especially in figure one, only one line is shown, so it’s not evident from the figure whether this line is representative of the system behaviour.

7)In figures 1, 2, and 3 it is difficult to see the numbers on the axes of the simulation images, and the axes are not labelled.

8)In figure 2, different colours correspond to different values of q, but it is not clarified which colour corresponds to which value of q.

9)The initial condition of the simulations does not seem to be mentioned. Does the outcome of the simulations depend on the initial condition?

These are cosmetic comments that the author may wish to consider, but that I believe are not necessary to be addressed:

10)There are two model versions used in the paper, ‘model 1’ and ‘model 2’. I believe the paper would be easier to read if the two model versions were named more descriptively. Maybe they could be ‘differentiation at division’ and ‘reversible differentiation’ (or ‘random’?)

11)Some of the paragraphs are really long and could be visually and topically split up more to make the paper easier to read.

**Have the authors made all data and (if applicable) computational code underlying the findings in their manuscript fully available?**

Reviewer #1: Yes

Reviewer #2: Yes

Reviewer #3: Yes

PLOS authors have the option to publish the peer review history of their article (what does this mean?). If published, this will include your full peer review and any attached files.

Reviewer #1: No

Reviewer #2: No

Reviewer #3: No
---

## [Decision Letter · Decision Letter 1]

19 Feb 2024

Dear Dr Greulich,

Thank you very much for submitting your manuscript "Emergent order in epithelial sheets by interplay of cell divisions and cell fate regulation" for consideration at PLOS Computational Biology.

As with all papers reviewed by the journal, your manuscript was reviewed by members of the editorial board and by several independent reviewers. In light of the reviews (below this email), we would like to invite the resubmission of a significantly-revised version that takes into account the reviewers' comments.

We cannot make any decision about publication until we have seen the revised manuscript and your response to the reviewers' comments. Your revised manuscript is also likely to be sent to reviewers for further evaluation.

Sincerely,

Philip K Maini

Academic Editor

PLOS Computational Biology

Jason Haugh

Section Editor

PLOS Computational Biology

Reviewer's Responses to Questions

**Comments to the Authors:**

Reviewer #1: The article "Emergent order in epithelial sheets by interplay of cell divisions and cell fate regulation" proposes a mechanism on cell fate regulation on epithelial sheets. The work uses an Ising model and couples cell fate and cell division.

The article is interesting, though not completely novel. Similar variants (and close relatives) of the Ising Model have previously considered similar interactions between cell fate and division.

Following my initial review, the authors have acknowledged the majority of my comments, except the work's relevance to the readers of PLoS Computational Biology, with the articles lack of data to validate, or even motivating the work. I do not accept that this would require significant funding as there is currently available data of sufficient quality within the literature.

I believe the work is of a high scientific quality, however, following on from my initial review, and the authors comments, I do not believe the article would be suitable for readers of PLoS Computational Biology. It is a heavily theoretical study, focusing mainly on analysing the theoretical model only. The authors do not relate their analysis to biological data, nor do they propose any real or relevant guidelines or directions for experimentalists to follow, driven by the work proposed here. I believe the work would be of interest to the scientific community, however, I believe it would appeal to those with a computational and theoretical understanding, in the work's current form. I would encourage the authors to pursue such avenues.

Reviewer #2: The author has accounted for some of the referees' comments and improved the manuscript. It is an interesting study, and I have the following suggestions:

(1) My concerns about the proximity of this paper to biological data and predictions remain. The author mentions very general experimental tests in the discussion, but the model is a toy model and not tied to a specific biological system. Since several referees brought up the proximity to biology as a concern, I would suggest adding biological images as motivation for the different types of qualitative cell arrangements in Figure 1 and pointing out what cells would represent cell type A and B in these images. I think the cartoon illustrations of different types of patterns that the author added are useful, but it would help link the model to biology more tightly if each of these illustrations was accompanied with a biological picture of a system the model could be meant to describe. Many biological papers contain images that can be reprinted for free with permissions through the Copyright Clearance Center.

(2) References to biological literature need to be added in some places. For example, at ‘These dynamics mimic, for example, cells in the basal layers of epidermis, oesohagus, or organotypic cultures, which are smooth sheets or have tubular geometry, and which may be regulated through juxtacrine Notch-Delta or Notch-Jagged signalling.’ at line 252 on pg. 14 could have references to review articles for these observations.

(3) I would suggest rewriting the sentence “The order parameter is the defining measure for phase transitions in complex systems, and is thus the gold standard to identify them.” The latter part (gold standard) is a strong statement and an opinion, so this should be clear. Pair correlation functions, order parameters, topological data analysis, radial distribution functions,… there are many choices for measuring the behavior of complex systems. I suggest adding a discussion with references to alternative quantitative approaches and a discussion of what they would add for future work (for example, the author mentions that the order parameter does not capture the variance of cluster sizes in their response, and this appears to be a feature of the simulations in Figure 3 that other quantitative approaches could capture).

Reviewer #3: The author has convincingly addressed my main concerns. In addition, the premise is more clearly stated, and the strengths and weaknesses of the paper are more openly discussed.

One of my main comments was a request to name possible biological systems to which the model could be applied. I asked about this since the author claims that the model suggests biological experiments. In the rebuttal, author mentions a reference (Jones et al 1995) which I thought was quite a nice way to address my comment. However, that reference does not appear to be cited in the revised manuscript. Surely, if the reference solidifies the premise of the paper, the readers should know about it?

**Have the authors made all data and (if applicable) computational code underlying the findings in their manuscript fully available?**

Reviewer #1: Yes

Reviewer #2: Yes

Reviewer #3: Yes

PLOS authors have the option to publish the peer review history of their article (what does this mean?). If published, this will include your full peer review and any attached files.

Reviewer #1: No

Reviewer #2: No

Reviewer #3: No
---

## [Decision Letter · Decision Letter 2]

5 Aug 2024

Dear Dr Greulich,

Thank you very much for submitting your manuscript "Emergent order in epithelial sheets by interplay of cell divisions and cell fate regulation" for consideration at PLOS Computational Biology. As with all papers reviewed by the journal, your manuscript was reviewed by members of the editorial board and by several independent reviewers. The reviewers appreciated the attention to an important topic. Based on the reviews, we are likely to accept this manuscript for publication, providing that you modify the manuscript according to the review recommendations.

Sincerely,

Philip K Maini

Academic Editor

PLOS Computational Biology

Jason Haugh

Section Editor

PLOS Computational Biology

Reviewer's Responses to Questions

**Comments to the Authors:**

Reviewer #2: The authors have accounted for my comments and I am happy to recommend acceptance. One small note: regarding "Persistent homology" in the parenthetical statement in the discussion, persistent should not be capitalized here.

Reviewer #4: The paper, 'Emergent order in epithelial sheets by interplay of cell divisions and cell fate regulation' uses a cellular automata model and theoretical insights from statistical physics to probe how the 'timing' of cell fate decisions, particularly with intercellular interactions, influences tissue structure. Overall, the paper is interesting and provides new insights into cell fate. However, I do echo previous reviewers' concerns around the fit for PLoS Computational Biology, as the manuscript is highly theoretical. (With apologies to the author for a) a late review that b) asks for even more work!)

Putting those concerns aside, my main comments are around clarity of the manuscript and increasing the biological relevance:

- The author investigates cell patterning with two cell types. Does the author have any insight into how the system would differ with more than two cell types (two spin states would no longer be sufficient to describe the system, for example), including where this may appear in biology? Could this be added to the discussion section?

- Is there any way to succinctly describe the method for how the neighbouring site j is chosen when a division event occurs? (This would be preferable to having to look up a referenced paper.)

- Am I correct in saying the in model C only p^\\lambda varies according to equations (5) and (6) and in model R only p^\\omega varies according to equations (5) and (6)? If so, could this please be made clearer in the text?

- Page 5, final paragraph: in ignoring the superscript for the probability functions, is the implication that the cell division and cell fate change events behave the same way for the different neighbouring cell configurations? Could this be either corrected or made clearer in the text?

- Page 7, line 137: The sentence starting with ‘For parameter’ is unclear - should this be ‘For the parameter J’? It should be made clear in the text that Fig 2 is for model C specifically.

- Sorry to be a pain, but are means and standard errors the most appropriate measures? Are the means and medians similar in your simulations, are the quantiles appropriately captured by the standard error? For example, in Fig 2. for the J=0.6, logistic example configuration, the configuration consists entirely of B cells, but this is not reflected in the top row plot of order parameter against signal strength. Apologies again for asking for 'more', but plots with medians and min/max or quantiles might better represent the sample simulations.

- For Fig 3, it might be worth noting the different J scales for the different models in the caption. A previous reviewer mentioned the ‘grey’ colouring in this figure (logistic, J = -0.3, q =0, Hill, J = -3.0, q = 0, J = -1.5, q = 0, for example). To assist readers, it may be worth noting that due to the resolution/magnification the ‘checkerboard’ patterning cannot be seen, or perhaps a zoomed in inset would be helpful?

- Page 9, top paragraph, would be good to have a summary (with biological interpretation) at the end, like the paragraph on Fig 2. Similarly, the second paragraph could use more biological insight. For example, for the parameter values in rows three and four of Fig 4., one cell type never dominates. The theoretical insights section would also be bolstered by biological interpretation.

- Fig 4: is it true that only the first two rows of Fig 4 show the logistic function on the left and the Hill function on the right? If so, could this be made clearer in the caption? (And if not, could the caption please be reworded?)

- Please excuse this comment if I have missed it, but what are p_3 and p_4 on page 14 line 229?

- Page 14: the assumption that probabilities do not depend on the cells neighbouring cell j are strong - can the implications of this assumption be discussed?

Minor (mainly typographical and admittedly petty, with apologies, especially if these comments come across as terse) comments:

- Fig 1. is slightly misleading given the model described in the paper has a square lattice, rather than a hexagonal lattice.

- Fig 1. Caption first sentence, I don’t think the word `respectively' is needed? Consistency: bottom row: or (Top Row)

- Order parameter figures: The captions state that \\phi is plotted with a bold line, but I believe both lines are bolded, I would suggest that the ‘label’ for the \\phi curve be ‘solid curve’.

- Page 2, line 35: what does ‘more or less’ mean? Is it needed?

- Page 2, line 50: please remove the word ‘are’.

- Page 3, line 72: comma not needed.

- Page 6, line 83: probability misspelt (currently spelt ‘probabilitiy’ - I am only stating this to help the author locate the misspelling).

- Page 7, line 140: the phrase ‘straight to’ is misleading, consider exchanging it for ‘rapidly to’.

- Page 12 after equation 11, Gillespie misspelt (currently spelt `Gillspie').

- Page 16, line 344: is the phrase ‘on the other hand’ warranted at the start of this sentence? The previous sentence has the same requirement that cell fate be reversible?

- The concluding paragraph/sentence of the paper is long and difficult to follow, I would consider re-writing.

**Have the authors made all data and (if applicable) computational code underlying the findings in their manuscript fully available?**

Reviewer #2: None

Reviewer #4: Yes

PLOS authors have the option to publish the peer review history of their article (what does this mean?). If published, this will include your full peer review and any attached files.

Reviewer #2: No

Reviewer #4: No

Figure Files:

Data Requirements:

Reproducibility:

References:

---

## [Decision Letter · Decision Letter 3]

6 Sep 2024

Dear Dr Greulich,

We are pleased to inform you that your manuscript 'Emergent order in epithelial sheets by interplay of cell divisions and cell fate regulation' has been provisionally accepted for publication in PLOS Computational Biology.

Best regards,

Philip K Maini

Academic Editor

PLOS Computational Biology

Jason Haugh

Section Editor

PLOS Computational Biology

Reviewer's Responses to Questions

**Comments to the Authors:**

Reviewer #4: The author has made some effort to address the reviewer comments. Therefore, I am content to see this paper accepted for publication. I commend the author on their efforts and persisting through multiple rounds of revisions.

**Have the authors made all data and (if applicable) computational code underlying the findings in their manuscript fully available?**

Reviewer #4: None

PLOS authors have the option to publish the peer review history of their article (what does this mean?). If published, this will include your full peer review and any attached files.

Reviewer #4: No

---

## [Editor Report · Acceptance letter]

4 Oct 2024

PCOMPBIOL-D-23-01212R3 

Emergent order in epithelial sheets by interplay of cell divisions and cell fate regulation

Dear Dr Greulich,

I am pleased to inform you that your manuscript has been formally accepted for publication in PLOS Computational Biology. Your manuscript is now with our production department and you will be notified of the publication date in due course.

With kind regards,

Anita Estes
